# Has Property Rights Reform of China's Farmland Water Facilities Improved Farmers' Irrigation Efficiency?—Evidence from a Typical Reform Pilot in China's Yunnan Province

**Yiyu Feng** [1] 🆔, **Ming Chang** [1], **Erga Luo** [2] **and Jing Liu** [1,*]

1   Institute of Agricultural Economics and Development, Chinese Academy of Agricultural Sciences, No. 12 Zhongguancun South Street, Haidian District, Beijing 100081, China
2   School of Public Administration, Zhejiang University, No. 866 Yuhangtang Road, Hangzhou 310013, China
*   Correspondence: liujing02@caas.cn; +86-13661322898

**Abstract:** China is implementing a pilot project to reform the property rights of farmland water facilities to solve the serious problems of chaotic end-of-pipe governance and inefficient irrigation from farmers. Based on microscopic research data of farm households in a typical pilot in Lu Liang County, Yunnan Province, in China, this study uses the Tobit model, and SEM was used to explore the impact of property rights reform on the irrigation efficiency of farmers and the potential mechanism paths. We further analyzed the differences in governance logic and irrigation efficiency between the two property rights models of "multiple cooperative governance" and "private contract governance" formed after the reform. The findings are as follows: (1) Compared with nonreformed areas, reformed areas have a higher promotion of adoption of water-saving technologies and a better quality of facility maintenance, which significantly contributes to irrigation efficiency, but farmers' perception of water scarcity negatively affects irrigation efficiency; (2) there are differences between the "multiple cooperative governance model" and "private contract governance model" in terms of the mechanism paths to improve irrigation efficiency, resulting in different focuses between the two models, in which the former one has better irrigation efficiency while cutting off some of the farmers' benefits and the latter benefits more farmers while losing some of the irrigation efficiency. Finally, this study recommends that China should continue to promote the reform of farmland water property rights while focusing on promoting water conservation technologies, improving the quality of facility maintenance and facility water supply capacity, enhancing farmers' awareness of water scarcity, and implementing a more water-efficient "multiple cooperative governance model". The results of this study provide a model with Chinese characteristics for developing property rights policies and governance models for farmland water facilities in developing countries.

**Keywords:** water facilities property rights; irrigation efficiency; multiple cooperative governance model; private contract governance model; Yunnan province; China

## 1. Introduction

China is currently facing problems such as unclear property rights, responsibilities, and chaotic end governance of farmland water facilities [1], leading to inefficient agricultural irrigation [2]. This has seriously affected China's food security, and thus farmland water reform is imperative. Since the abolition of the "agricultural tax" (China fully abolished the agricultural fee tax as of 1 January 2006) and the "two-worker system" (the "two-work system" refers to the employment system that requires villagers to provide labor for the public welfare of the village without compensation, specifically divided into "compulsory labor" and "labor accumulation labor"), village collectives have lost the economic basis and motivation to organize farmers for farmland water construction and directly deploy farmers' labor for village farmland water construction and maintenance. Thus, small farmland water facilities present someone to use but no one to manage, resulting in their

aging and dilapidated state [3,4], which has led to more than half of the water used in agriculture being wasted during water transportation [5]. Therefore, under the current new situation and challenges of China's rural revitalization, a severe form of food security threats, and a serious waste of agricultural water, there is an urgent need to reform the property rights of farmland water facilities (farmland water facilities in this study refer to the headworks, trunk canals, branch canals, bucket canals and agricultural canals that connect rivers and reservoirs for agricultural irrigation services) and establish an efficient small farmland water governance model, which has become a national strategic need to alleviate China's agricultural water shortage and improve irrigation efficiency.

China has launched a series of policy reforms and explorations on farmland water management, such as the establishment of "water user associations (WUA), water rights trading reforms, and comprehensive agricultural water price reforms". Although these policies have been effective in improving the governance of China's farmland water resources and promoting irrigation efficiency [6], they still fail to address the fundamental problems of chaotic governance of China's farmland water facilities and a severe waste of water resources [7]. Some scholars have found that many WUAs exist only in form and do not play a fundamental role [8,9]; the water rights trading system also has many problems in practical application [10]; at the same time, the promotion of agricultural water price reform policy also faces many difficulties [11,12].

There has been numerous academic research on farmland water management policies which has provided the research basis for this paper. However, most studies focused on the dimension of "water" and ignored the perspective of "property rights of agricultural water facilities". Therefore, in 2014, China introduced the policy of "launching a pilot project of reforming the property rights system and innovating the operation and management mechanism of farmland water facilities", which aims to solve the current problems of grassroots governance for farmland water facilities and improve irrigation efficiency by clarifying the property rights of farmland water facilities and exploring various governance models.

Regarding the property rights of farmland water facilities and the irrigation efficiency of farmers, scholars have conducted extensive research on both from different perspectives. Ostrom (1990) [13] argued that farmland water facilities are typical rural public pond resources, and Coase (1960) [14] believed that clarifying their property rights could produce better stewardship of public pond resources. On this basis, transferring management to farmers' associations or other private sectors effectively solves rural water management problems [15], and has been successful in most countries [16]. On the contrary, in the absence of clear property rights, each person in the irrigated community makes decisions on resources utilization in their own interest, resulting in excessive and uncontrolled use and causes severe negative externalities [17,18]. In addition, some researchers have argued that contracting rural public goods such as farmland and water resources to private parties to form a "market contracting system" is more likely to achieve service specialized and effective governance [19]. In particular, scholars have further found that clear property rights of farmland water facilities are the basis for irrigation water tariff setting, which in turn is the primary determinant of farmers' water-saving behavior [20]. However, Woubet (2018) [21] suggested that the ability of any external agency policy to be effectively understood by the recipient is an issue that must be considered. Therefore, the property rights reform of China's farmland water facilities must focus on the farmer level. For example, a series of problems (e.g., low efficiency of agricultural irrigation, serious waste of water resources, and insufficient adoption of water-saving technologies in China for a long time) are closely related to the low cost of agricultural water paid by farmers [22–26], while the adoption of water-saving technologies will contribute to the improvement of irrigation efficiency [27,28]. At the same time, farmers' perception of water scarcity directly shapes water use behavior and indirectly affects irrigation efficiency [29]. In addition, differences in tariff-setting rules and water-using rules can also lead to differences in irrigation efficiency, and studies have found that "quota management water use" performs better than "metered water use" in improving water-saving efficiency [30,31]. Therefore,

from the farmer's perspective, we should focus on the above mentioned factors that can affect the irrigation efficiency.

A comparative analysis of existing studies shows that little research was conducted on the direct impact of property rights reform of farmland water facilities on farmers' irrigation efficiency, and successful experiences on such kind of reform are also still rarely presented. Then, China's pilot exploration of property rights reform of farmland water facilities provides an excellent opportunity for observation, for which we propose the following questions: Has this policy affected China's farmland water governance? Has it improved the irrigation efficiency of farmers? What kind of governance model and institutional rules have emerged? Therefore, this paper attempts to answer the above questions and fill the gaps in previous studies by conducting a study in a typical reform pilot in Yunnan, China, analyzing the impact of property rights reform of farmland water facilities on farmers' irrigation efficiency, exploring its impact mechanisms and presenting successful governance models. It provides empirical evidence to alleviate water scarcity and promote farmland water governance.

## 2. Policy Background and Impact Mechanism Analysis Framework

In this section, we first sort out and summarize the policies of property rights reform of farmland water facilities in China and the promotion of the reform pilot. On this basis, we select a typical case pilot to analyze its specific system and propose a theoretical analysis framework of the impact of property rights reform of farmland water facilities on farmers' irrigation efficiency based on the analysis of the typical pilot combined with existing studies.

### 2.1. China's Farmland Water Facilities Property Rights Reform, and Policy Promotion

2.1.1. Policy Sorting and Reform Promotion

In 2011, the No. 1 document of the Central Government proposed the "Decision on Accelerating the Reform and Development of Water Resources", kicked off the reform of China's farmland water resources (Figure 1), which was the first time in 62 years since the founding of the People's Republic of China that the central government document made a comprehensive deployment of water resources work, which laid the foundation for the subsequent reform of property rights of farmland water facilities. By 2014, China's Ministry of Water Resources, Ministry of Finance, and Development and Reform Commission jointly implemented the" reform of the property rights system for farmland water facilities" in 100 pilot counties across the country, which focused on clarifying and transferring project property rights, innovating the operation and management model, and promoting water conservation. Since then, China has introduced a series of reform policies for farmland water resources, all of which involve improving and promoting the property rights system for farmland water facilities.

2.1.2. The Effectiveness of the Reform in the 100 Pilot Counties

The 100 pilot reform lasted from 2014 to 2018 ended and entered the deepening reform and model promotion stage (Figure 1). This paper summarizes and analyzes the government research report and the compiled information on the acceptance and inseption materials of the 100 pilot reform counties and finds that the pilot reform has achieved six general achievements, namely, clarifying the property rights of facilities, promoting good operation and maintenance of facilities, promoting water conservation, improving farmers' income, and establishing a large number of specialized and socialized irrigation service teams.

At the same time, each pilot region has formed its unique model; in fact, there are mainly two property rights models: the first is the form of property rights integration, that is, ownership, management, revenue, and supervision under one subject; the second is the form of property rights separation, namely, ownership, management, revenue, and supervision under two or more subjects.

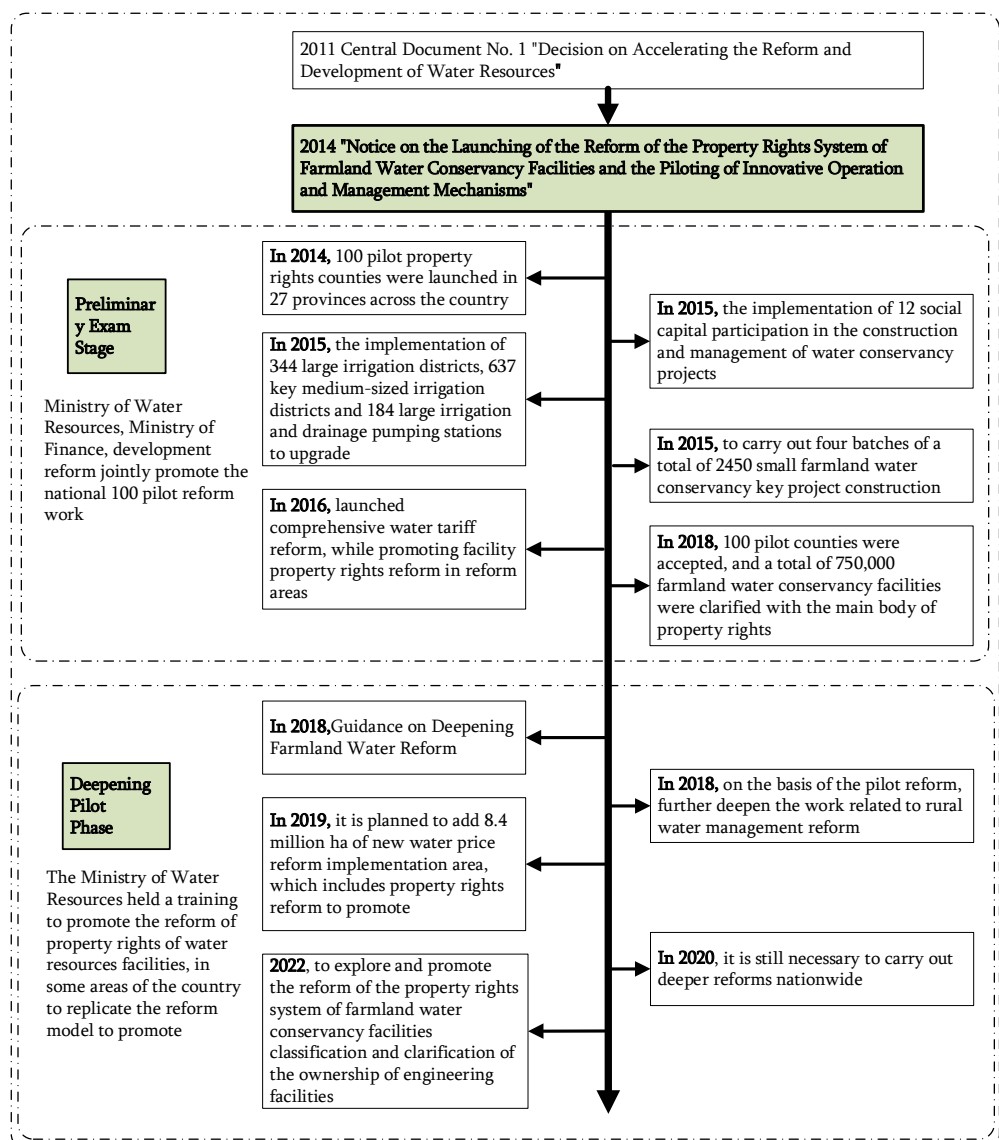

**Figure 1.** Organizational approach used to promote changes to China's policies on farmland water property rights as a means to improve agricultural water conservation.

*2.2. Reform in Lu Liang County, Yunnan, China—An Analysis of the Institutional Presentation and Governance Model of a Typical Pilot Case*

2.2.1. Typical Case Pilot Selection and System Presentation

This study adopts the principle of theoretical sampling, which requires the selection of typical regions that are suitable to answer the research questions [32]. This paper focuses on how farmland water property rights reform affects farmers' irrigation efficiency, with special emphasis on the deeper exploration of the "what" and "how" questions [32,33]. Lu Liang County is the only county among these 100 counties that has formed two property rights models and is the most successful county in terms of reform. Thus, the 2018 field meeting to promote the governance model of property rights reform of China's farmland water facilities was held here and attended by Vice Premier Hu Chunhua, making the Yunnan Lu Liang reform pilot an ideal typical case to answer those research questions.

Since 2014, the reform pilot in Lu Liang County, Yunnan Province, has been carried out mainly in Zhongba Village and Chaotie Village in Xiaobaihu Township. The common approach to reform in both villages is to clarify the property rights of farmland water facilities (ownership, management, revenue, supervision) and to develop a definite water pricing system, management system, and irrigation system formulated based on clear

property rights. However, these two neighboring villages have spontaneously formed two different property rights models, and both of them have achieved remarkable results. The specific forms of property rights and institutional rules of these two models are very different, and this study summarizes them as the "multiple cooperative governance model"(MCG-Model) and "private contract governance model" (PCG-Model) according to the differences between these two forms of property rights (Table 1).

**Table 1.** Brief description of the differences between the two property rights models in Lu Liang County, Yunnan, China.

| Model Name | Zhongba Village "Multiple Cooperative Governance" Model (MCG-Model) | Chaotie Village "Private Contract Governance" Model (PCG-Model) |
|---|---|---|
| Grassroots governance organizations | **Village Committee + Cooperative + Subdistrict Water Steward** | **Cooperative + Contractor** |
| Reform time | 2014 | 2014 |
| Property Rights System | The ownership, operation, revenue, and supervision right belongs to the village collective | The property right of the project and the supervision right belongs to the cooperative; the operation and revenue right belongs to the contractor. |
| Water Use System | "Water quota management": total water consumption control, charge by Mu(1Mu=0.07Hectare) (ranging from 200–250 RMB/0.07Hectare according to the difficulty of water distribution in the plot) | "Measured water price": uniform water price for the whole village, 0.79 RMB/m$^3$ |
| Management system | Multiple combination management systems: the village committee unified leadership, entrusted to the professional cooperatives to manage. The cooperative organizes farmers to democratically elect water managers in each district to maintain irrigation facilities and carry out irrigation and water distribution for farmers, while the cooperative pays the water managers' salaries and supervises their work. | Private contract management: The contractor of the project operation right is fully responsible for the operation and maintenance of the irrigation facilities. |
| Irrigation system | Unified centralized water release irrigation system: the unified frequency and timing of water release by the village committee, 7–12 times a season. | On-demand water supply system: release water at any time according to the needs of water users, unlimited times and volume. The contractor provides complete irrigation services. |
| Differences in water-saving technology adoption | The universal use of drip irrigation | The universal use of sprinkler irrigation |

### 2.2.2. Governance Logic of the "Two Property Rights Model" in Typical Case Pilot

Through our long-term field research and in-depth interviews, and summary analysis of the case areas, we found that the governance logic of the two property rights models is as follows:

1. The governance logic of the "multiple cooperative governance" model

Governance Logic (Figure 2): The MCG-Model is a top-down governance mode of "village authority + cooperative + village elites+ farmers" under clear property rights. The village collective owns the property rights of the project, the water cooperative acts as an agent for the village collective to execute the operation rights, and the "village elites" democratically elected by the farmers are the actual governance subjects as subdistrict water stewards. The motivation of the water controllers comes mainly from the salary incentive and performance assessment of the parent cooperative, while the convenience of the farmers' water use, the timeliness of water deliveries, and the maintenance of facilities are the criteria for the performance assessment of the water stewards. In this model, village

collective organizations have an absolute say. To facilitate unified management, promote fair water use, and reduce farmers' planting risks, village collective organizations have played the role of rational economic agents. The main manifestations of this are: the establishment of a uniform irrigation system with a fixed frequency and time limit of water releases each year; the implementation of a water tariff system with charges based on planting area; and the uniform promotion of drip irrigation technology, which ultimately promotes water conservation among farmers.

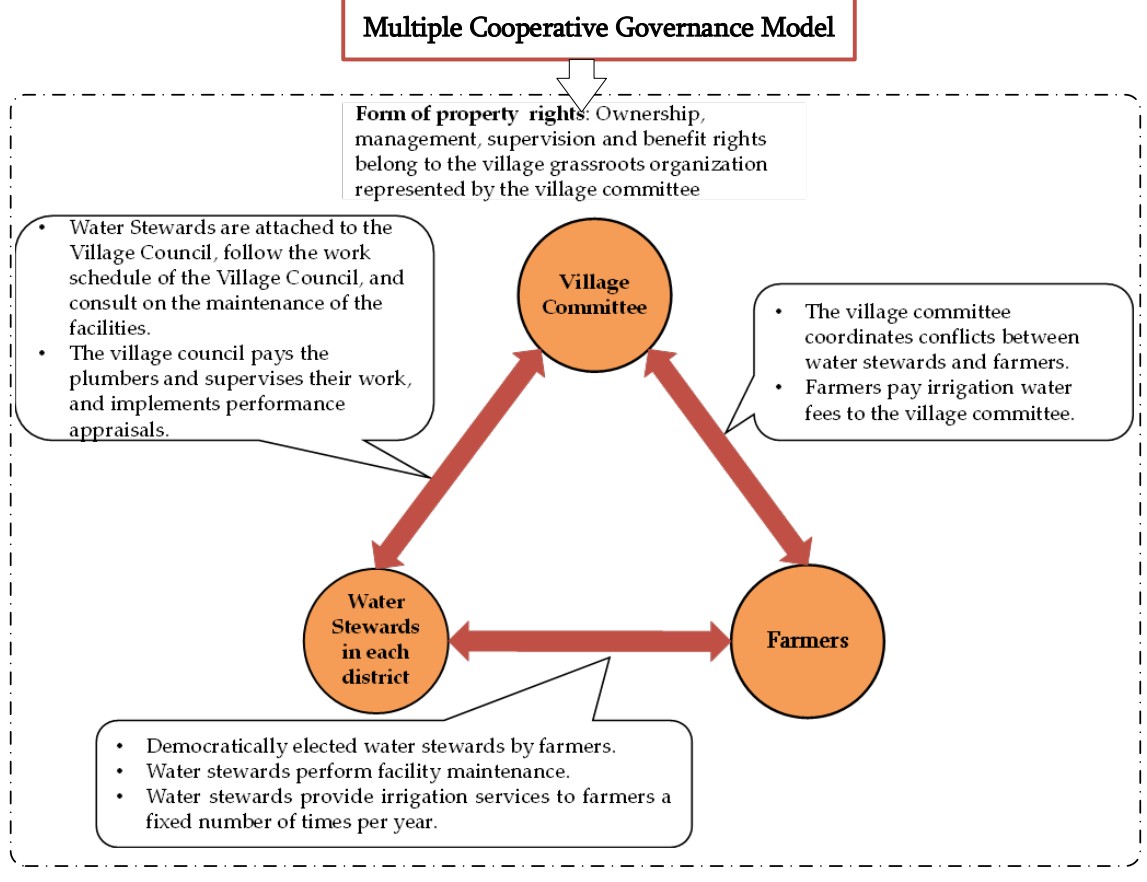

**Figure 2.** Intrinsic governance logic diagram of the multiple governance model.

2.     Governance logic of the "private contract governance" model

Governance logic (Figure 3): The "private contract governance" model shows a bottom-up governance approach of "market contracting" under the privatization of management rights. In this model, the agricultural water cooperative (WUA) has the ownership of the project, the contractor has the operating right, and is the only responsible body for the management of farmland water facilities, which is supervised by WUA. The contractor has the power to make rules. In the process of making rules, he plays the role of a rational economic person whose interest is motivated by the collection of water charges. As the contractor's income is determined by the amount of water used by the farmers, he wants the farmers to use as much water as possible, so the contractor makes the water tariff system of metered water and the irrigation system of demand-based irrigation, but the contractor has no right to interfere with the farmer to adopt drip irrigation or sprinkler irrigation. In fact, the contractor does not want the farmer to adopt more water-saving irrigation technology. While farmers play the role of rational economists in adopting water-saving technologies, they choose sprinkler irrigation technology to minimize installation costs. Therefore, the private contracting model actually creates an upward transmission of farmers' water demand, and the contractors are motivated by the benefits to improve

governance, take the initiative to maintain irrigation facilities, improve their water supply capacity, and provide market-based irrigation services. As a result, farmers have achieved water savings.

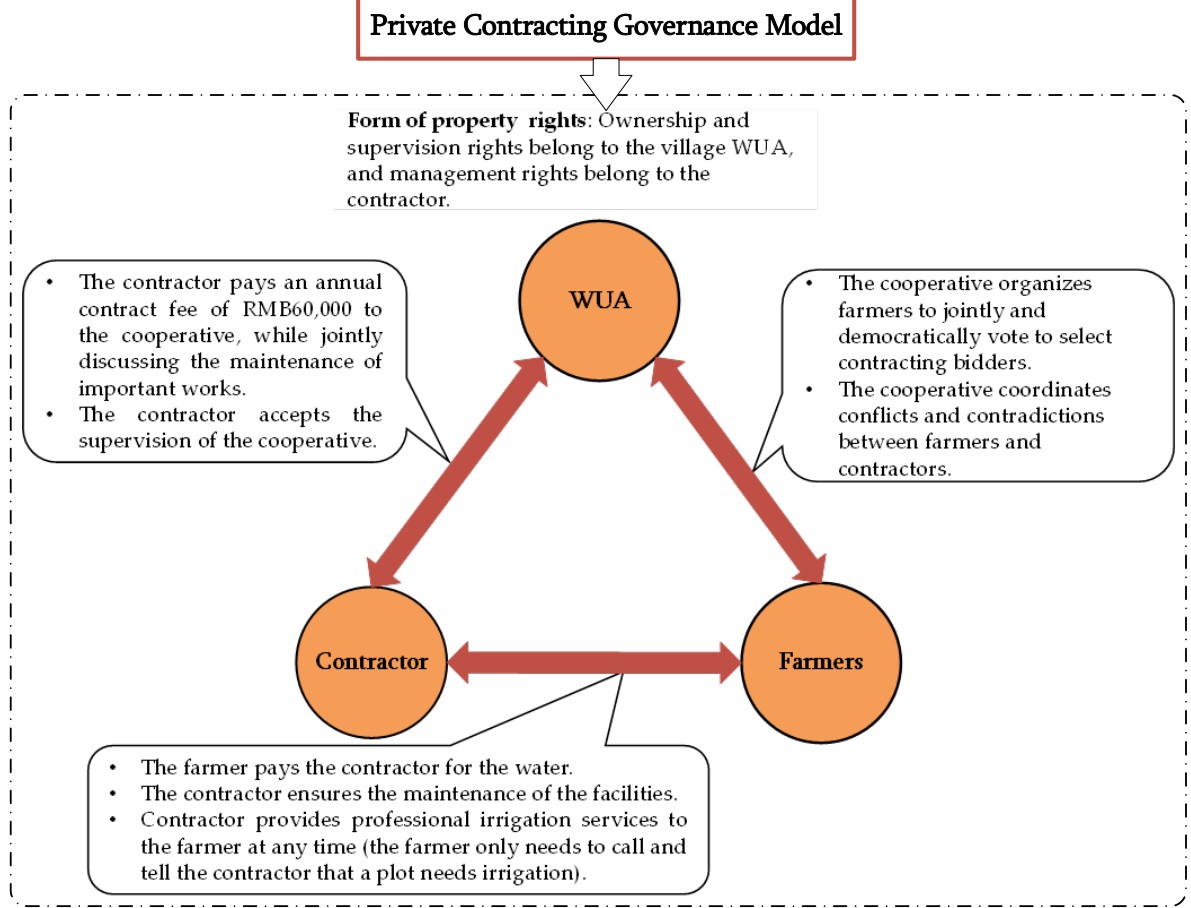

**Figure 3.** Governance logic inherited in the private contracting governance model.

Based on extensive field interviews with local farmers, government officials, and water bureau personnel, this paper presents the institutional cases of the abovementioned typical reform pilots and analyzes the internal logic of the two property rights governance models. However, can the conclusions drawn from experience and case summaries be effectively verified empirically? Whether the property rights reform indeed promote farmers' water conservation or not? What are the differences between the two models in terms of farmers' water savings? Based on the above case studies, we propose the following research hypothesis and analytical framework for further empirical validation.

### 2.3. A Theoretical Mechanism Analysis Framework Based on a Typical Model

According to the above questions and the literature analysis, this study proposes an analytical framework to assess of the impact of property rights reform on irrigation efficiency (Figure 4).

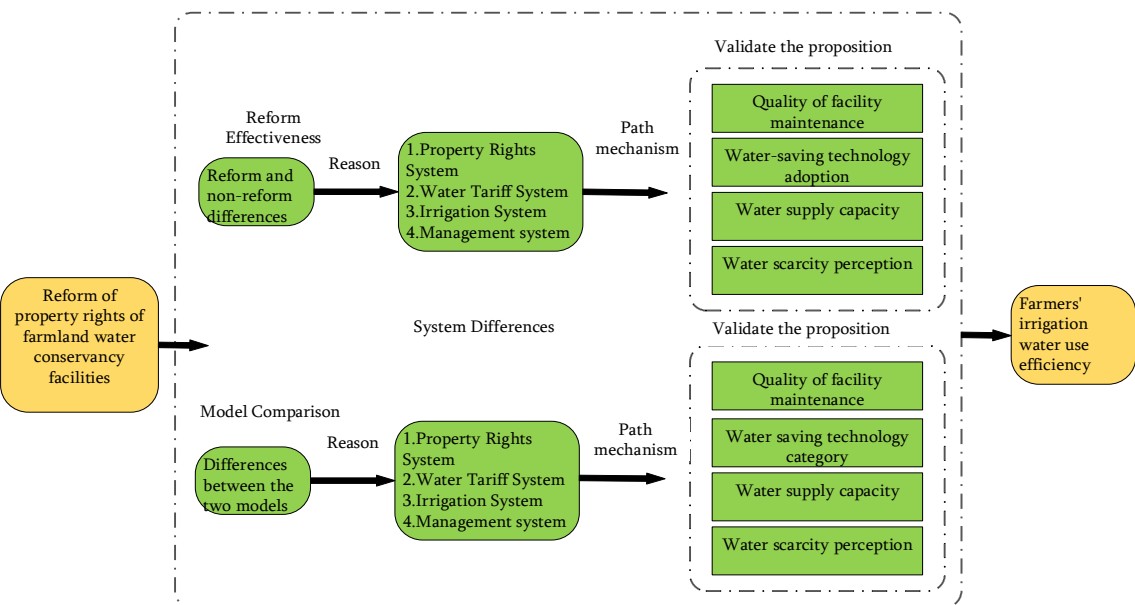

**Figure 4.** Theoretical mechanism framework of the impact of property rights reform of farmland water facilities on the irrigation efficiency of farming households.

Ostrom (1990) [13] proposed eight basic principles of smallholder water governance through some typical case studies, arguing that the "long-term autonomous governance" of smallholder water resources can be achieved on the basis of rulemaking. After the property rights of farmland water facilities were clarified in the reform area of Lu Liang, Yunnan, a series of basic institutional rules were formulated. Rulemaking is likely to affect farmers' irrigation efficiency through different mechanistic pathways. With this in mind, the specific mechanism pathway analysis framework in this paper is:

1.  On the one hand, tariff setting makes farmland water facilities a profitable private product, and property owners, as rational economists, will maintain farmland water facilities to obtain continuous income from water charges collection, which is likely to improve the quality of facility maintenance and thus promote irrigation efficiency [34].

2.  On the other hand, after the water tariff setting, farmers are also likely to choose to adopt water-saving technologies to save irrigation costs under the assumption of rational economic man, thus promoting irrigation efficiency [18,26].

3.  Burt (1997) [35] argued that the effectiveness of an irrigation system is mainly reflected in the timeliness and reliability of its water delivery and transportation. Hence, the development of a clear irrigation and stewardship system is likely to reduce delays in the delivery of water from the facility and enhance the water supply capacity, thus contributing to the efficiency of irrigation.

4.  Finally, a series of institutional developments following property rights reform may affect farmers' perceptions of water scarcity, and farmers' perceptions of water scarcity are likely to affect irrigation efficiency [36].

5.  Following property rights reform, differences in the form of property rights and institutional rules between the "MCG-Model" and "PCG-Model" may lead to differences in the quality of facility maintenance, the type of water-saving technologies employed, the water supply capacity of the facilities, and farmers' perceptions of the extent of water scarcity. In turn, these differences are likely to lead to differences in irrigation efficiency between the two models. Some studies concluded that irrigation is more efficient under a water quota management system than under a metered charge system and that water saving is more efficient with drip irrigation than with sprinkler irrigation [12,31]. The two models in this study are precisely different in water use systems and in water conservation technology adoption, so the "MCG-Model" is likely to result in higher irrigation efficiency than the "PCG-Model".

Based on the above analysis of theory and reality, this study proposes the following research hypothesis:

**Hypothesis 1.** *The reformed areas have higher irrigation efficiency than the nonreformed areas.*

**Hypothesis 2.** *Property rights reform of farmland water facilities affects farmer irrigation efficiency by impacting the quality of irrigation facility maintenance, facility water supply capacity, farmers' water conservation technology adoption, and farmers' perception of water scarcity.*

**Hypothesis 3.** *The "MCG-Model" results in higher irrigation efficiency than the "PCG-Model".*

**Hypothesis 4.** *Differences in the internal institutional logic between the "MCG-Model" and the "PCG-Model" lead to differences in the quality of facility maintenance, capacity of water supplying facility, farmers' adoption of water conservation techniques, and farmers' perceptions of water scarcity, ultimately leading to differences in the impact of each pathway variable on irrigation efficiency.*

## 3. Data and Methods

### 3.1. Data Source

The data were mainly obtained from field investigations conducted by our research team at Luliang County in December 2020, June 2021, and July 2021. We conducted in-depth field interviews and farmer questionnaires in two reformed villages and four randomly selected nonreformed villages during the field research. Finally, 345 valid farmer questionnaires were collected, with a valid rate of 98.3%. Finally, we selected the most important local irrigation crop, "spring potato", as the research object and obtained data from 328 farmers after excluding farmers who did not grow spring potato, including 208 in the reform area (108 in the MCG-Model and 100 in the PCG-Model) and 120 in the control area. To avoid the effect of extreme values, all continuous variables are winsorized at 1% above and below [12]. Finally, we apply STATA15 and AMOS 20 for empirical analysis.

### 3.2. Variables and Descriptive Statistics

#### 3.2.1. Dependent Variable

Nominal irrigation efficiency: estimated irrigation water use efficiency of the largest plot of each farmer growing spring potatoes based on a superefficient data envelopment analysis (DEA) model. The current assessment of irrigation efficiency is primarily based on the traditional DEA method [37–40] and superefficient DEA methods [41,42]. The main purpose of this study is to measure the difference in efficiency between the samples; however, the problem of simultaneous validity of several decision units arises when estimating irrigation efficiency using traditional DEA methods (efficiency equal to 1), which makes further comparisons impossible and thus affects the accuracy of the parameters. While the superefficient DEA model can avoid the drawback that the decision units are simultaneously one and cannot be compared [43–45], the superefficient DEA model is used in this paper to avoid this drawback.

$$min\theta \; s.t. \begin{cases} \sum_{\substack{k=1 \\ k\neq i}}^{N} X_k\lambda_k + S^- = \theta X_i \\ \sum_{\substack{k=1 \\ k\neq i}}^{N} Y_k\lambda_k - S^+ = Y_i \\ \lambda_k \geq 0, k = 1,2,\dots,N \\ S^+ \geq 0, S^- \geq 0 \end{cases} \quad (1)$$

In Equation (1) $\theta$ is the target planning value, $\lambda_k$ ($k = 1, 2, \dots, N$) is the planning decision variable, and $S^-$, $S^+$ are the slack variable vectors.

Irrigation water use efficiency is defined as the ratio of the optimal amount of inputs to the actual value of irrigation water, combined with the actual input elements of the study, as shown in Table 2, with the following formula:

$$WUEi = \frac{AWRi - Sw, i}{AWRi} \tag{2}$$

In Equation (2), $WUEi$ is the irrigation water use efficiency of farmer $i$; $AWRi$ is the actual irrigation water input of farmer $i$; $Sw, i$ is the slack variable vector in irrigation water input of farmer $i$ calculated by the superefficient DEA model, and $AWRi$ minus $Sw, i$ is the optimal irrigation water input.

In this study, the results of irrigation efficiency were measured according to the above method, and the descriptive statistics (Table 3) of the average irrigation efficiency for the full sample, reformed areas sample, and non-reformed areas sample were 0.41, 0.49, and 0.29, respectively, and the average irrigation efficiency of the reformed areas was significantly higher than that of the non-reformed areas. The average irrigation efficiency of the "MCG-Model" was 0.51, which was higher than that of the "PCG-Model," which was 0.47.

**Table 2.** Input-output indicator system for measuring irrigation efficiency of farmers' largest plots for spring potato growing.

| Indicator Type | Indicator Name |
| --- | --- |
| Inputs | Amount of seed input (500 g/0.07Hectare) |
| | Amount of machinery input (RMB/0.07Hectare) |
| | Fertilizer input quantity (500 g/0.07Hectare) |
| | Amount of pesticide weeding input ((RMB/0.07Hectare) |
| | Amount of labor input (standard labor day/0.07Hectare) |
| | Amount of agricultural film input (500 g/0.07Hectare) |
| | Amount of irrigation water input (m$^3$/0.07Hectare) |
| Output | Spring potato yield (500 g/0.07Hectare) |

### 3.2.2. Core Independent Variables

Core independent variables: property rights reform of farmland water facilities, property rights development model. Among them, the property rights reform of farmland water facilities is the core independent variable for testing Hypotheses 1 and 2; the property rights development model is the core independent variable for testing Hypotheses 3 and 4.

Farmland water facilities property rights reform: The core content has been given in the theoretical section. In this study, farmland water facilities property rights reform is for 0–1 variables, 1 represents the reform village, and 0 represents the control group. Property rights development model: The specific model has been given in the theoretical description section; 1 represents the MCG-Model, and 0 represents the PCG-Model.

### 3.2.3. Path Variables

Path variables: water-saving irrigation technology adoption behavior (WAST), quality of irrigation facility maintenance, water supply capacity, and farmers' perception of water scarcity. Among them, WAST is a 0–1 variable; irrigation facility maintenance quality is an ordered multi-categorical variable, from 1–5 representing facility maintenance quality from low to high; the water supply capacity variable is measured by the number of irrigation delays for farmers, with more delays representing weaker water supply capacity; farmers' water scarcity perception is a 0–1 variable, with 1 representing farmers' perception of future water scarcity and 0 representing the perception of no water scarcity.

**Table 3.** Descriptive statistics of variables.

| Variable Name | Variable Meaning and Assignment | All Samples (Mean) | Reform Zone (Mean A) | Nonreform Zone (Mean B) | MCG-Model (Mean C) | PCG-Model (Mean D) |
|---|---|---|---|---|---|---|
| Dependent variables | | | | | | |
| Irrigation efficiency | Continuous variables | 0.42 | 0.49 | 0.29 | 0.51 | 0.47 |
| Core independent variables | | | | | | |
| Reform of property rights of farmland water facilities | Farmland water facilities property rights reform area = 1; nonreform area = 0 | 0.63 | 1 | 0 | 1 | 1 |
| Property rights development model | MCG-Model = 1; PCG-Model = 0 | 0.33 | 0.52 | 0 | 1 | 0 |
| Path variables | | | | | | |
| WAST | Adopting water-saving irrigation technology = 1; not adopting= 0 | 0.76 | 0.95 | 0.43 | 0.94 | 0.97 |
| WAST category | Sprinkler irrigation = 1; Drip irrigation = 2; Not used = 0 | 1.03 | 1.39 | 0.40 | 1.82 | 1.03 |
| Quality of facility maintenance | 1 = very poor; 2poor; 3 = fair; 4 = good; 5 = very good | 3.43 | 3.86 | 2.68 | 3.81 | 3.90 |
| Water supply capacity | How many times was your potato irrigation delayed last year? (Number of times) | 1.01 | 0.49 | 1.92 | 0.93 | 0.12 |
| Water scarcity perception | Future shortage of water for agricultural irrigation in this village = 1; no shortage = 0 | 0.53 | 0.44 | 0.68 | 0.60 | 0.31 |
| Control variables | | | | | | |
| Total household income | Continuous variable (Ten thousand RMB) | 13.84 | 15.00 | 11.00 | 12.48 | 17.81 |
| Planting scale | Continuous variable (0.07Hectare) | 1.51 | 1.65 | 1.27 | 1.14 | 2.09 |
| Distance to the county | Continuous variable (kilometers) | 15.09 | 14.82 | 15.55 | 14.44 | 15.14 |
| Age | Continuous variable (years) | 48.58 | 47.13 | 51.11 | 47.97 | 46.42 |
| Education level | Continuous variable (years) | 7.60 | 7.67 | 7.47 | 7.33 | 7.97 |
| Whether village cadres | Yes = 1; 0 = No | 0.11 | 0.10 | 0.13 | 0.10 | 0.11 |
| Farming experience | Continuous variable (years) | 27.61 | 26.27 | 29.94 | 26.71 | 25.90 |
| Largest plot irrigation condition | 1 = very poor; 2 = poorly; 3 = fairly; 4 = better; 5 = very good | 3.73 | 4.04 | 3.19 | 3.85 | 4.20 |
| Maximum plot quality | 1 = first class; 2 = second class; 3. third class; 4. equal field | 1.91 | 2.01 | 1.72 | 1.94 | 2.08 |
| Topographical conditions of the largest plot | 1 = very uneven; 2 = sloping; 3 = more flat; 4 = very flat | 1.25 | 1.33 | 1.10 | 1.43 | 1.25 |
| Distance of the largest plot from the dam | Continuous variable (km) | 3.70 | 3.25 | 4.50 | 2.46 | 3.91 |
| Irrigation frequency | Continuous variable (times) | 7.80 | 8.32 | 6.91 | 7.46 | 9.04 |

Table 3 shows that the average quality of facility maintenance in the full sample, reformed area, non-reformed area, MCG-Model, and PCG-Model is 3.43, 3.86, 2.68, 3.81, and 3.90, respectively. The average WAST in the full sample, reformed area, non-reformed area, MCG-Model, and PCG-Model is 0.76, 0.95, 0.43, 0.94, and 0.97, respectively. The average water supply capacity (number of irrigation delays) in the full sample, reformed area, non-reformed area, MCG-Model, and PCG-Model is 1.01, 0.49, 1.92, 0.93, and 0.12, respectively. The average farmers' perception of water scarcity in the full sample, reformed area, non-reformed area, MCG-Model, and PCG-Model is 0.53, 0.44, 0.68, 0.60, and 0.31, respectively.

### 3.2.4. Control Variables

Control variables: farmer household characteristics, individual farmer characteristics, and plot characteristics. In this study, with reference to other related studies, the control variables were selected to include household characteristics, individual household characteristics, and plot characteristics [12,36]. Among them, household characteristics mainly include total annual household income, the area of the farming operation, and the distance of the household from the county town. The personal characteristics of farmers mainly include education, age, farming experience, and whether they are village cadres. Plot characteristics in this study include soil quality, irrigation conditions, topographic conditions, distance from dams, and irrigation frequency in the largest plots where farmers grow spring potatoes. Descriptive statistics of the variables are shown in Table 3.

### 3.3. Model Setting

First, the Tobit multiple linear regression model was used to test whether farmland water reform affects farmers' irrigation efficiency and to construct a PSM (propensity score matching method) counterfactual framework for result validation. Second, the Tobit multiple regression model was also used to test the differences between the two models affecting farmers' irrigation efficiency. Third, structural equation path analysis was used to study the mechanism path of farmland water reform affecting farmers' irrigation efficiency. Fourth, structural equation path analysis was also used to analyze the differences between the mechanism path of the two models affecting farmers' irrigation efficiency.

### 3.3.1. Tobit model

Considering that the value of the irrigation efficiency results measured by the super-efficient DEA model ranges from a minimum value of 0 to a maximum value that may exceed 1, showing a left broken-tail distribution at 0, it is a limited dependent variable. If OLS regression is performed, it will make the measurement results inaccurate, and the Tobit model is suitable for the regression analysis of the limited dependent variable in this study [46]. Therefore, in conjunction with the analytical framework, the benchmark model for the multiple regression of the property rights reform of farmland water facilities on the irrigation efficiency of farmers is set as follows.

Model 1: Regression of farmland water facility property rights reform on farmers' irrigation efficiency

$$WUE_i = \alpha_0 + \alpha_1 IR_i + \alpha_2 X_i + \varepsilon_i \tag{3}$$

where $WUE_i$ denotes farmer irrigation efficiency, $IR_i$ is the core independent variable (whether in the reform zone), $X_i$ represents the remaining control variables, and $\varepsilon_i$ is the random disturbance term. $\alpha_0$ is the constant term, and the other parameters are the regression coefficients.

Model 2: Regression of the property rights development pattern on the irrigation efficiency of farm households

$$WUE_i = \beta_0 + \beta_1 MGM_i + \beta_2 H_i + \delta_i \tag{4}$$

where $WUE_i$ denotes farmer irrigation efficiency, $MGM_i$ is the core independent variable (property rights development model), $H_i$ represents the remaining control variables, and

$\delta_i$ is the random disturbance term. $\beta_0$ is the constant term, and the other parameters are the regression coefficients.

### 3.3.2. PSM Counterfactual Matching and Equation Estimation

Since Tobit multiple linear regression can only estimate the conditional expectation of the core explanatory variables on the dependent variable and is susceptible to sample selection bias, it is likely to interfere with the truthfulness of the results. The problem of sample bias between reformed and nonreformed villages is likely to exist in this study. Since the data in this study are cross-sectional, the commonly used DID method is not applicable to this study. Therefore, the propensity score matching method (PSM) is an effective analytical method to address the above problem, the basic idea of which is to construct a "counterfactual" framework to eliminate sample selection bias by finding counterfactual control groups similar to the treatment group [47]. The steps of PSM counterfactual framework analysis include estimating propensity scores, matching method selection, common support hypothesis testing, balance testing, and measuring estimated average treatment effects. In this regard, the logit model was applied to estimate the propensity score value. Accordingly, the decision equation of whether the sample villages chose to implement the reform of property rights of farmland water facilities is as follows:

$$P(E_i) = P(IR_i = 1 | E_i) = \frac{\exp(\beta E_i)}{[1 + \exp(\beta E_i)]} \tag{5}$$

In Equation (5), the two terms on the left-hand side are the probability that the farmer is in the reformed zone given $E_i$. The right-hand side represents the cumulative distribution function, and $\beta$ is the coefficient of the characteristic variable.

In this paper, assuming that $WUE_{1i}$ is the irrigation efficiency index of farmers in the treatment group, $WUE_{0i}$ is the irrigation efficiency index of farmers in the control group, and $IR_i$ is the treatment variable, the average treatment effect (ATT) of the effect of property rights reform on irrigation efficiency can be expressed as:

$$\text{ATT} = \text{E}(WUE_{1i} | IR_i = 1) - \text{E}(WUE_{0i} | IR_i = 1) = \text{E}(WUE_{1i} - WUE_{0i} | IR_i = 1) \tag{6}$$

With the help of existing studies [12,48], "distance from the farmer to the county town, topographic condition of the largest plot, distance from the largest plot to the dam, years of education, total business area, and the total number of irrigations" were selected as matching variables in this study. In addition, to ensure the robustness of the regression results, the nearest neighbor matching method, radius matching method, kernel matching, and local linear regression matching methods were used to test the robustness of the PSM model.

### 3.3.3. Structural Equation Path Analysis Model

In the test of the mechanism path of how the property rights reform of farmland water facilities affected irrigation efficiency, the test of multiple mechanism path variables involved in the theoretical framework of the study was considered. While the structural equation path model (SEM) can analyze the relationship between multiple independent variables, multiple dependent variables, and multiple mediating variables at the same time [49,50], this study established a path analysis model of how property rights reform affects irrigation efficiency, with the following basic expressions.

$$\eta = B\eta + \Gamma\xi + \zeta \tag{7}$$

In Equation (7), $\xi$ is the exogenous variable matrix; $\eta$ is the endogenous variable matrix; $B$ and $\Gamma$ are both structural coefficient matrices, and $B$ denotes the interaction between the components of the endogenous variable matrix $\eta$, $\Gamma$ represents the effect of $\xi$ on $\eta$; $\zeta$ is the residual matrix. Based on the above basic expressions, the path analysis

model of the reform of property rights of farmland water facilities affecting the irrigation efficiency of farmers in this study was established as follows:

Model 3: Path analysis model

$$WUEi = \alpha_1 IR_i + \varepsilon_1$$

$$WST_i = \alpha_2 IR_i + \varepsilon_2$$

$$FM_i = \alpha_3 IR_i + \varepsilon_3$$

$$WSC_i = \alpha_4 IR_i + \varepsilon_4$$

$$USEP_i = \alpha_5 IR_i + \varepsilon_5$$

$$WUE_i = \alpha_0 + \alpha_1 IR_i + \alpha_2 WST_i + FM_i + \alpha_4 WSC_i + +\alpha_5 USEP_i + \varepsilon_i \tag{8}$$

where $WUEi$ represents irrigation efficiency, $IR_i$ represents core explanatory variables, $WST_i$ represents WAST behavior, $FM_i$ represents irrigation facility maintenance quantity, $WSC_i$ represents water supply capacity, $USEP_i$ represents farmers' perception of water scarcity, $\varepsilon$ is a random disturbance term, $\beta_0$ is a constant term, and other parameters are regression coefficients. The path analysis model of the effect of the two models on irrigation efficiency is similar to the above model and will not be repeated here.

## 4. Results

Before analyzing how the property rights reform of farmland water facilities affects the irrigation efficiency of farm households, it is necessary to determine whether the property rights reform of farmland water facilities affects the irrigation efficiency of farm households. Therefore, the first part of this section will use multiple regression analysis to test whether the property rights reform of farmland water facilities affects irrigation efficiency. The second section uses structural equation path analysis to verify the influence mechanism.

### 4.1. Empirical Results

4.1.1. Impact of Property Rights Reform on the Irrigation Efficiency of Farm Households

1. Baseline model results, sample regression results after PSM matching

Table 4 reports the estimated results of farmland water facilities' property rights reform affecting irrigation efficiency under the full sample. Among them, Model 1 includes only the core independent variables, Model 2, Model 3, and Model 4 are based on Model 1 with the addition of household characteristics, individual characteristics, and maximum plot characteristics, respectively, and Model 5 is the sample estimation effect after PSM matching, and Model 6 is the robustness test after replacing the estimation method based on Model 5. Comparing the estimation results of models (1–4), it can be seen that the coefficients of the effects of the core explanatory variables are significantly positive whether the control variables are added individually or simultaneously; therefore, it can be tentatively judged that the reform of the property rights of farmland water facilities has significantly improved the irrigation efficiency of farmers. Meanwhile, after gradually controlling for household characteristics, personal characteristics, and plot characteristics, the overall explanatory power of the model improved by more than 5%. According to model 5, the estimation results after PSM matching prove that the above conclusions are robust, and model 6 further ensures the reliability of the results. All the above analysis results prove that the reform of the property rights of farmland water facilities has a significant positive impact on the irrigation efficiency of farmers and that hypothesis 1 is valid.

**Table 4.** Empirical analysis of the impact of property rights reform of farmland water facilities on the irrigation efficiency of farm households.

| Model | Tobit | | | | PSM-Tobit | PSM-OLS |
|---|---|---|---|---|---|---|
| | Model 1 | Model 2 | Model 3 | Model 4 | Model 5 | Model 6 |
| Explained variables | Irrigation efficiency | | | | | |
| $IR_i$ | 0.198 *** | 0.168 *** | 0.170 *** | 0.164 *** | 0.213 *** | 0.213 *** |
| | (7.52) | (6.33) | (6.32) | (5.27) | (−7.22) | (−7.05) |
| Household characteristics | — | Controlled | Controlled | Controlled | Controlled | Controlled |
| Individual characteristics | — | — | Controlled | Controlled | Controlled | Controlled |
| Plot characteristics | — | — | — | Controlled | Controlled | Controlled |
| Constant term | 0.290 *** | −0.493 *** | −0.506 *** | −0.419 *** | −0.239 | −0.239 |
| | (13.81) | (−2.28) | (−2.13) | (−1.68) | (−1.02) | (−0.99) |
| N | 328 | 328 | 328 | 328 | 302 | 302 |
| LR | 52.12 *** | 73.4 *** | 80.89 *** | 85.8 *** | 85.58 *** | |
| $R^2$ | | | | | | 0.23 |

Note: Robust standard errors are in parentheses; *** represent significance at the 1% statistical level.

2.   Common support test

This paper uses the logit model to estimate the propensity score for setting up a farmland water facilities property rights reform village. To ensure the reasonableness and validity of the PSM estimation, the following common support hypothesis test is conducted. Taking the nearest neighbor matching method within the caliper as an example, Figure 5 shows that after PSM matching, the kernel density functions of the propensity scores of the treatment and control groups are relatively close, and the propensity score values of both groups mostly fall within a common range of values, indicating a high quality of matching.

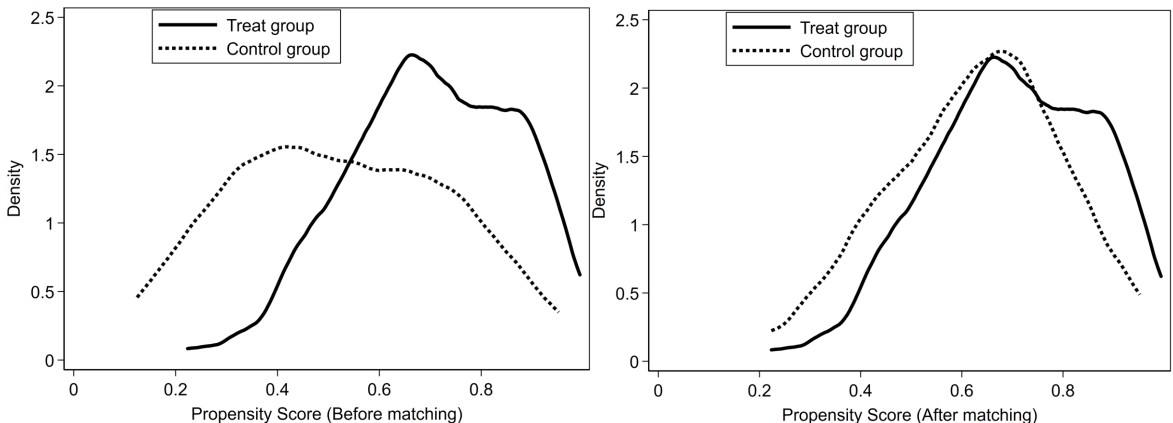

**Figure 5.** Kernel density function plot before and after propensity score matching.

3.   Balance test

To ensure the balance of the samples after matching, this paper uses k-nearest neighbor matching, k-nearest neighbor matching within the caliper, radius matching, kernel matching, and local linear regression matching for matching. The results of the balanced test in Table 5 show that both the Pseudo R2 and LR chi 2 decreased significantly after matching compared to those before matching, while the deviations of the means were all less than 10%. The median deviation decreased from 34.1 to 6.2~9.1. Based on the results of the above analysis, it is clear that the PSM model used in this paper significantly reduces

the sample selection bias and passes the balance test, **implying that the data matching is well balanced.**

**Table 5.** Balance test results.

| Matching Method | Pseudo R2 | LR chi2 | *p* Value | Mean Bias | Median Bias |
|---|---|---|---|---|---|
| Before matching | 0.14 | 60.03 | 0.000 | 34.1 | 33.6 |
| k-nearest neighbor matching (n = 1) | 0.018 | 5.35 | 0.000 | 8.8 | 6.7 |
| Caliper matching (r = 0.02) | 0.009 | 5.03 | 0.000 | 9.1 | 8.9 |
| k-nearest neighbor matching within caliper (r = 0.02, n = 1) | 0.004 | 2.05 | 0.000 | 5.9 | 5.8 |
| Kernel matching | 0.012 | 6.78 | 0.000 | 9 | 4.1 |
| Local linear regression matching | 0.004 | 2.42 | 0.000 | 6.2 | 6 |

## 4. Estimation of the Average Treatment Effect

Table 6 shows that the estimation results obtained in this study using five different matching methods are generally consistent, indicating good robustness of the propensity score matching results. After PSM estimation, it is found that the property rights reform of farmland water facilities has a significant positive effect on farmers' irrigation efficiency, and Hypothesis 1 is further verified. Specifically, the mean treatment effect is 0.199, indicating that after resolving the sample selectivity bias, the reform of the property rights of farmland water facilities led to a significant increase of 19.9% in the irrigation efficiency of farm households.

**Table 6.** Average treatment effect of propensity score matching (ATT).

| Matching Method | Irrigation Efficiency |
|---|---|
| k-nearest neighbor matching (n = 1) | 0.188 *** (0.030) |
| Caliper matching (r = 0.02) | 0.202 *** (0.028) |
| k-nearest-neighbor matching within caliper (r = 0.02, n = 1) | 0.193 *** (0.029) |
| Kernel matching | 0.207 *** (0.027) |
| Local linear regression matching | 0.207 *** (0.030) |
| Mean value | 0.199 |

Note: Robust standard errors are in parentheses; *** represents significance at the 1% statistical level.

4.1.2. Differences in the Effects of Different Property Rights Models on Farmers' Irrigation Efficiency

Table 7 reports the estimated results of two specific property rights models affecting irrigation efficiency in the reformed areas under the reduced sample scenario (core independent variables: MCG-Model = 1; PCG-Model = 0). The model settings are the same as in the previous section. Model 5 is the robustness test after replacing the estimation method. The estimation results show that models 1, 2, and 3 are not significant. With the inclusion of all control variables in model 4, the core explanatory variables are significantly positive at the 10% level, and the robustness test of model 5 proves the robustness of the estimation results. The above analysis shows that there are differences in irrigation efficiency between the two property rights models, and the irrigation efficiency of farmers in the "MCG-Model" is relatively better than that in the "PCG-Model", proving that hypothesis 3 holds.

### 4.2. Path Mechanism Analysis

The above multiple linear regression analysis led to two conclusions: (1) the property rights reform of farmland water facilities in Luliang County, Yunnan Province, China, significantly improved the irrigation efficiency of farmers; (2) among the two different property rights models, the "MCG-Model" was better than the "PCG-Model" in terms of the irrigation efficiency of farmers.

However, what are the mechanisms through which the property rights reform of farmland water facilities has improved irrigation efficiency for farmers? What are the differences in institutional pathways between the two models that lead to better irrigation efficiency in the "MCG-Model" than in the "PCG-Model"? The specific mechanism path is further verified in the following section.

**Table 7.** An empirical analysis of the differences in the effects of different property rights models on the irrigation efficiency of farm households.

| Model | Model 1 | Model 2 | Model 3 | Model 4 | Model 5 |
|---|---|---|---|---|---|
| | Tobit | Tobit | Tobit | Tobit | OLS |
| Explained variables | Irrigation efficiency | | | | |
| | 0.037 | 0.058 | 0.058 | 0.067 * | 0.067 * |
| $MGM_i$ | (1.05) | (1.59) | (1.62) | (1.71) | (1.65) |
| Household characteristics | — | Controlled | Controlled | Controlled | Controlled |
| Individual characteristics | — | — | Controlled | Controlled | Controlled |
| Plot characteristics | — | — | — | Controlled | Controlled |
| Constant term | 0.471 *** | −0.480 | −0.521 | −0.572 | −0.572 |
| | (19.65) | (−1.33) | (−1.39) | (−1.46) | (−1.41) |
| N | 208 | 208 | 208 | 208 | 208 |
| LR | 1.10 | 14.18 *** | 23.17 *** | 26.57 *** | |
| R2 | | | | | 0.12 |

**Note:** Robust standard errors are in parentheses; ***, * represent significance at the 1%, and 10% statistical levels, respectively.

4.2.1. Path Mechanism Analysis of Farmland Water Facilities Property Rights Reform Affecting Irrigation Efficiency of Farm Households

1. Model fitness evaluation

Based on the theoretical analysis framework and Equation (8), this section tests theoretical Hypothesis 2, i.e., whether the property rights reform of farmland water facilities contributes to farmers' irrigation efficiency through four theoretical mechanisms: "farmers' adoption of water-saving technologies, quality of irrigation facility maintenance, farmers' perception of future water scarcity, and water supply capacity".

In the process of model construction, this paper deletes variables based on the consideration of whether there is a theoretical relationship and whether the path of influence is significant and combines the correction index to amend the model of the mechanism path of the property rights reform of farmland water facilities affecting irrigation efficiency until it meets the criteria related to the evaluation of model fit [50]. Table 8 presents the overall structural equation model fitness evaluation indicators, judgment criteria, and fitness results. The results show that important indicators such as RMSEA, AGFI, and CFI pass the test, indicating that the theoretical model fits the sample data relatively well and that the estimation results have high reliability.

**Table 8.** Path analysis model fit indices of the reform of property rights of farmland water facilities affecting irrigation efficiency.

| Evaluation Indicators | Criteria or Thresholds for Adaptation | Fitted Value | Whether to Adapt to Determine |
|---|---|---|---|
| Cardinality ($\chi^2$) | $p > 0.05$ | $p = 0.00$ | No |
| Asymptotic residual mean square and root square (RMSEA) | <0.09 | 0.086 | Yes |
| Goodness-of-fit index (GFI) | >0.90 | 0.986 | Yes |
| Adjusted Adequacy Index (AGFI) | >0.90 | 0.938 | Yes |

2.    Estimation results

The results of the structural path model estimation are shown in Figure 6, and the results of the analysis of the specific path regression coefficients are shown in Table 9. According to the results of the path analysis, the reform of the property rights of farmland water facilities promotes irrigation efficiency by promoting farmers' adoption of water conservation techniques and improving the quality of facility maintenance but reduces farmers' perception of agricultural water shortages, leading to a decrease in irrigation efficiency, and the water supply capacity variable is not significant. From the results of the above study, it can be obtained that the remaining mechanisms in Hypothesis 3 have been verified except for the water supply capacity path.

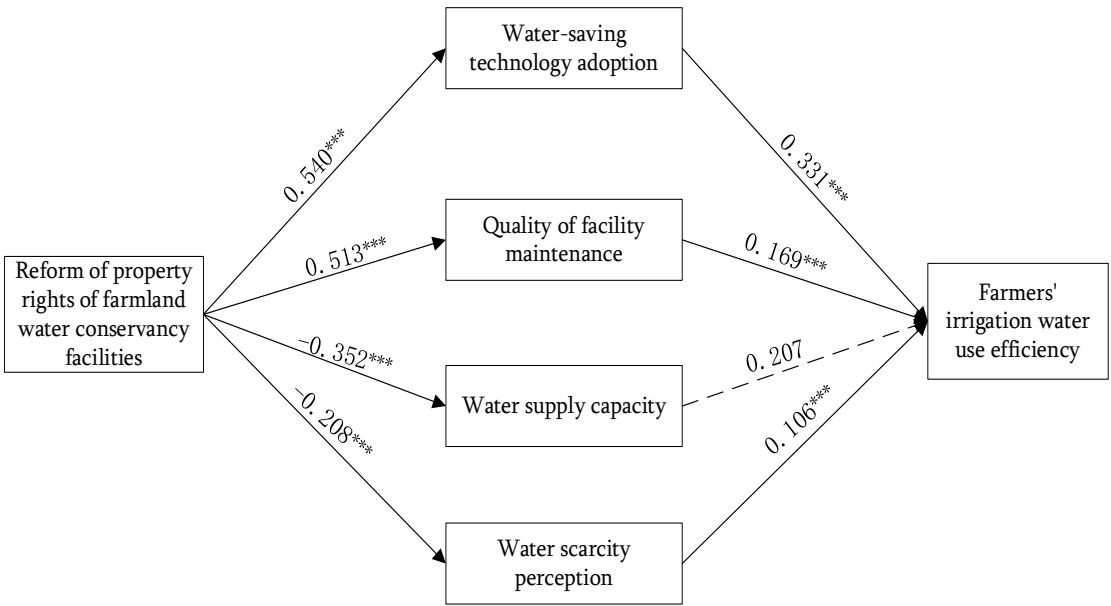

**Figure 6.** Pathways of the effect of property rights reform of farmland water facilities on irrigation efficiency. Note: Solid arrows indicate significant paths of action, and dashed arrows indicate insignificant paths of action; values next to arrows are path coefficients; *** represent significance at the 1% statistical level.

**Table 9.** Standardized regression coefficient results of the path analysis of the impact of property rights reform on the irrigation efficiency of farmland water facilities.

| Paths | Standardized Estimated Coefficients | Standard Deviation | Threshold Ratio Value | *p* Value |
|---|---|---|---|---|
| Property rights reform—>water-saving technology adoption | 0.54 | 0.047 | 11.601 | 0.000 |
| Property rights reform—>Maintenance quality | 0.513 | 0.111 | 10.807 | 0.000 |
| Property rights reform—>water supply capacity | −0.352 | 0.214 | −6.807 | 0.000 |
| Clear property rights—>water scarcity perception | −0.208 | 0.056 | −3.851 | 0.000 |
| Water-saving technology adoption—>Irrigation efficiency | 0.331 | 0.027 | 6.206 | 0.000 |
| Maintenance quality—>Irrigation efficiency | 0.169 | 0.012 | 3.014 | 0.003 |
| Water supply capacity—>Irrigation efficiency | −0.027 | 0.007 | −0.487 | 0.626 |

3.    Mechanism path analysis

Table 10 shows the calculated results of the indirect and total effects of the path coefficients for each mechanism. Overall, the combined effect of the mechanism paths increased irrigation efficiency by 0.244 standard units in the reformed zone relative to the nonreformed zone. The indirect effects of each mechanism pathway are analyzed specifically below.

**Table 10.** Indirect and total effects of property rights reform of farmland water facilities on irrigation efficiency.

| Impact Path | Indirect Effects | Percentage of Contribution |
|---|---|---|
| Property rights reform—>water saving technology adoption—>irrigation efficiency | 0.179 | 73.36% |
| Property rights reform—>Maintenance quality—>Irrigation efficiency | 0.087 | 35.66% |
| Property rights reform—>perception of water scarcity—>irrigation efficiency | −0.022 | −9.01% |
| Total effect | 0.244 | 100% |

Path 1: Water-saving technology adoption path

The property rights reform—>water saving technology adoption—>irrigation efficiency path is significantly positive. From Table 10, we can see that the adoption of water-saving technologies after the property rights reform improved the irrigation efficiency of farmers by 0.179 standard units, with a contribution of 73.36%. Through in-depth interviews with farmers, it was found that the reform in Yunnan Province formulated the agricultural water tariff system after the clarification of project property rights, which changed the previous zero-cost form of agricultural water use, and the agricultural water price increased the production cost of farmers, and farmers in the reform area gradually adopted water-saving irrigation technology to reduce the cost of water use. Therefore, adopting water-saving technology promoted the improvement of irrigation efficiency of agrarian households [26].

Path 2: Facility maintenance quality path

The path of property rights reform—>facility maintenance quality—>irrigation efficiency is significantly positive, which means that the improvement in the quality of maintenance of the facilities after the reform contributed to an increase of 0.087 standard units of irrigation efficiency, with a contribution of 35.66% (Table 10). Through in-depth interviews with property owners, we found that in reformed areas where property rights are clear, charging for water means that the property owner has an endogenous incentive to maintain the facility. At the same time, the implementation of the field project management responsibility system has also prompted the owner of the operation rights to maintain and repair the irrigation facilities on time, which effectively ensures the good operation of the irrigation facilities, reduces the waste of leakage in the process of lifting and transporting water for agricultural irrigation, and ultimately promotes the improvement of farmers' irrigation efficiency, which is the same as the conclusion of the study already conducted by Chang (2022) [12].

Path 3: Farmers' water scarcity perception path

The property rights reform—>water scarcity perception—>irrigation efficiency path is significantly negative, which shows that the property rights reform has instead reduced farmers' awareness of water scarcity, leading to a decrease in irrigation efficiency by 0.022 standard units, with a contribution of 35.66% (Table 10). Originally, this was contrary to common sense inferences and to existing research [36], but in-depth interviews with farmers revealed that it is not difficult to explain because existing research on farmers' perceptions of water scarcity is not premised on property rights reform of water facilities. However, in this study, it is mainly because the improvement of various systems after the property rights reform has made farmers' irrigation water guaranteed, thus raising their psychological expectation of more adequate agricultural water resources in the future, and farmers feel that there is no water shortage, so they may use water resources wastefully, resulting in lower irrigation efficiency. The findings of this study remind us even more to pay attention to improving farmers' awareness of water conservation and education in the process of property rights reform because the reform of farmland water facilities does not solve the current situation of insufficient total water resources.

Path 4: The water supply capacity path is not significant

The path of property rights reform—>water supply capacity—>irrigation efficiency path is insignificant. Through in-depth interviews with farmers, it was found that the

number of delays in irrigation was the indicator of water supply capacity used in this study, and the number of delays in irrigation was significantly reduced in the reformed areas after the property rights reform. However, farmers in nonreformed areas will use their initiative to irrigate their crops on time by pulling water with trucks, pumping water with diesel engines, and pumping water with wells, which ultimately results in little difference in the number of irrigation delays with the reformed area. However, this is exactly the problem that China needs to solve now, and it is also where the reform comes in. Farmers in the nonreform areas in this study spent many human and material resources under the same number of irrigation delays as in the reform areas, and although the water supply capacity variable here appears to be insignificant, there is actually a huge difference in water supply conditions between the reform and nonreform areas. On the other hand, it also indicates that the "number of irrigation delays" to measure water supply capacity in this study tends to be insignificant under different irrigation conditions, and perhaps other researchers can further find better "water supply capacity" variables under different irrigation conditions to explore this path.

### 4.2.2. Path Analysis of the Impact of Different Property Rights Development Models on Farmers' Irrigation Efficiency

1. Model fitness evaluation

This section explores what mechanism paths lead to higher irrigation efficiency in the MCG-Model than in the PCG-Model. Next, based on theoretical Hypothesis 4, we further analyze the differences in the impact of the type of water-saving irrigation technology adopted by farmers, farmers' perceptions of water scarcity, quality of facility maintenance, and water supply capacity on irrigation efficiency under the two property rights models.

The model correction process is the same as in the previous section. The results of the fit indices of the model are shown in Table 11, and the important indices of the model fit indices RMSEA, AGFI, and CFI all passed the test, indicating that the theoretical model fits the sample data relatively well and that the estimation results have high reliability.

**Table 11.** Fitted indices of the path analysis model of the property rights development pattern affecting irrigation efficiency.

| Evaluation Indicators | Criteria or Thresholds for Adaptation | Fitted Value | Whether to Adapt to Determine |
|---|---|---|---|
| Cardinality ($\chi^2$) | $p > 0.05$ | $p = 0.00$ | No |
| Asymptotic residual mean square and root square (RMSEA) | <0.09 | 0.050 | Yes |
| Goodness-of-fit index (GFI) | >0.90 | 0.991 | Yes |
| Adjusted Adequacy Index (AGFI) | >0.90 | 0.950 | yes |

2. Estimation results

The estimation results are shown in Figure 7, and the specific path regression analysis is presented in Table 12. Based on the results of the path analysis, it can be seen that the differences in irrigation efficiency between the two models are indirectly caused by the differences in the two path variables of farmers' water conservation technology adoption category and water supply capacity. The adoption of drip irrigation technology in the MCG model leads to a better increase in irrigation efficiency than the PCG-Model, but the greater water supply capacity in the PCG-Model leads to a better increase in irrigation efficiency than the MCG model. The quality of facility maintenance and farmers' perception of water scarcity variables are insignificant.

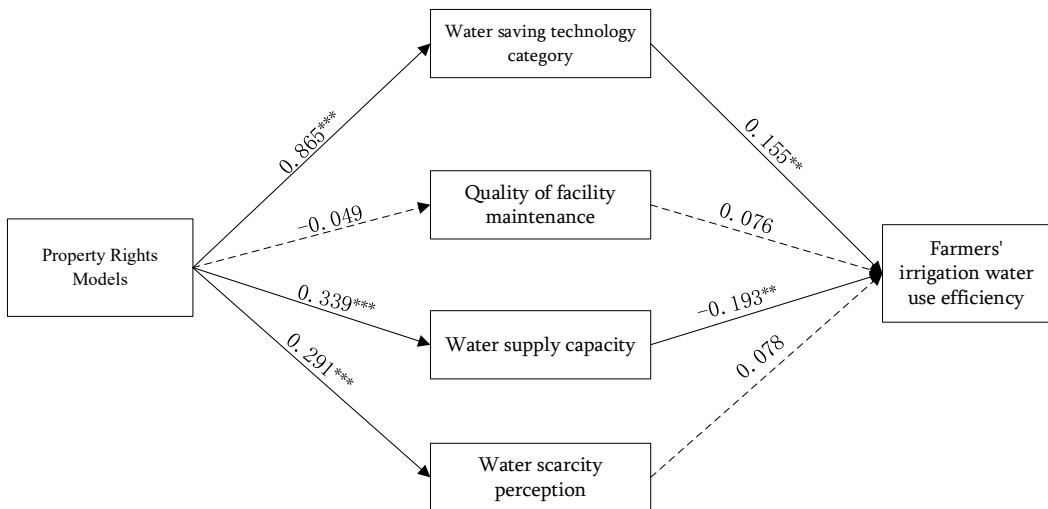

**Figure 7.** The path of action of the property rights development model on the impact of irrigation efficiency. ***, ** represent significance at the 1%, and 5% statistical levels, respectively.

**Table 12.** Standardized regression coefficient results for path analysis of the impact of property rights development patterns on irrigation efficiency.

| Path | Standardized Estimated Coefficients | Standard Deviation | Threshold Ratio Value | *p* Value |
|---|---|---|---|---|
| Property Rights Models—>Water-saving technology categories | 0.865 | 0.035 | 24.753 | 0.000 |
| Property Rights Models—>Maintenance Quality | −0.049 | 0.129 | −0.713 | 0.476 |
| Property Rights Models—>Water supply capacity | 0.339 | 0.157 | 5.181 | 0.000 |
| Property Rights Models—>perceived water scarcity | 0.291 | 0.066 | 4.378 | 0.000 |
| Water saving technology category—>Irrigation efficiency | 0.155 | 0.037 | 2.133 | 0.030 |
| Maintenance quality—>Irrigation efficiency | 0.076 | 0.02 | 1.04 | 0.298 |
| Water supply capacity—>Irrigation efficiency | −0.193 | 0.016 | −2.572 | 0.010 |
| Water scarcity perception—>Irrigation efficiency | 0.078 | 0.037 | 1.086 | 0.278 |

3.  Mechanism path analysis

Table 13 shows the indirect and total effects of the path coefficients of each mechanism variable obtained through the calculations. In terms of the total effect, the irrigation efficiency of the MCG-Model is 0.069 standard units higher than that of the PCG-Model. The following section analyzes the indirect effects of each mechanism path specifically.

**Table 13.** Indirect and total effects of MCG-Model on irrigation efficiency effects.

| Impact Path | Indirect Effects | Percentage of Contribution |
|---|---|---|
| Property Rights Models—>Water saving technology adoption categories—> irrigation efficiency | 0.134 | 194.20% |
| Property Rights Models—>water supply capacity—>irrigation efficiency | −0.065 | −94.20% |
| Total Effect | 0.069 | 100% |

Path 1: Water-saving technology adoption category differences path

The path of the Property Rights Models—>Water saving technology adoption categories—>irrigation efficiency is significantly positive, which means that the widespread adoption of drip irrigation technology by farmers under the "MCG-Model" leads to 0.134 standard units higher irrigation efficiency than the adoption of sprinkler irrigation technology by farmers

in the "PCG-Model", with a contribution of 194.20%. (Table 13). This is in line with existing studies, where drip irrigation is more water-efficient than sprinkler irrigation [12]. Statistics show that 88% of the farmers in the "MCG-Model" have adopted drip irrigation technology. In contrast, 95% of the farmers in the private contract model generally use sprinkler irrigation. However, why do the two property rights models lead to differences in the types of technology adoption by different farmers? The main reason for this phenomenon, which has been analyzed in the previous section of this study, is that whoever has the right to profit and operate the irrigation facilities has the power to set the rules, and the rules ultimately lead to differences in the adoption of water-saving technologies by farmers.

Path 2: Water supply capacity path

The Property Rights Models—>water supply capacity—>irrigation efficiency path is significantly negative, which means that the irrigation facilities of the "MCG-Model" have more delays, resulting in a decrease in irrigation efficiency of 0.065 standard units compared to the "private contract governance", with a contribution of −94.20% (Table 13). In-depth interviews with farmers and business rights owners revealed the following reasons: In the "PCG-Model", a metered water tariff system and a demand-based irrigation system are in place, and the contractors want to maximize their income, hoping that farmers will use as much water as possible without delaying every drop of water they need and ensuring that irrigation facilities are open at all times, so the contractors are motivated by profit to minimize the number of delays in irrigation and maximize the water supply capacity of the irrigation facilities. However, in the "MCG-Model", the system of water quota management and charging for water according to the size of the planted area results in fixed income for the cooperatives, so providing irrigation services to farmers at any time does not increase the cooperative's income; therefore, the cooperatives do not have the endogenous motivation to provide timely irrigation services to farmers. At the same time, because the fixed number of releases and fixed time of the year to irrigate farmers do not take into account the regularity of crop water needs and whether each plot and crop of farmers is at the time when they need irrigation the most, it may lead to irrigation delays. This ultimately leads to relatively low irrigation efficiency for farmers in the MCG-Model under the water supply capacity path.

Paths 3 and 4: Facility maintenance quality and water scarcity perception paths

The effects of the quality of facility maintenance and the farmers' perception of water scarcity on irrigation efficiency are not significant. The reason for this is that, despite the differences in the internal governance logic of the two models, both models reach a superior state in terms of facility maintenance quality and do not differ significantly in their effects on farmers' perceptions of water scarcity, which ultimately leads to a nonsignificant effect of the two mediating factors on farmers' irrigation efficiency.

## 5. Further Discussion

### 5.1. Discussion on the Impact of Property Rights Reform of Farmland Water Facilities on the Irrigation Efficiency of Farm Households

The results of this study demonstrate that the property rights reform of farmland water facilities in Yunnan, China, significantly improved the irrigation efficiency of farmers. The results of studies on the improvement of water use efficiency by clear property rights are consistent with the findings of existing studies [51,52]. In addition, the present study is unique and innovative when compared with existing studies. Although studies have also demonstrated that the allocation of property rights can promote water efficiency, most water property studies have focused on "water rights". For example, Rong (2013) [51] found that water resource rights can improve water use efficiency. Gao (2021) [52] believed that water rights trading can improve the efficiency of water use. Most of the research is based on the "water" level, such as water rights, water rights allocation, and water rights trading, to study the impact on agricultural irrigation efficiency, neglecting the means of transportation of "agricultural water", i.e.; ignoring the study of water use efficiency based on the property rights of farmland water facilities. Studies on the property rights

of farmland water facilities have also mainly focused on the impact of property rights on the maintenance and governance of facilities [18,26], and the impact on enhancing the collective action capacity of farmers [1]. Therefore, the contribution of this study is to: (1) assess the improvement of irrigation efficiency by the property rights reform of farmland water facilities, (2) propose specific mechanisms and paths of the impact of property rights reform of farmland water facilities on irrigation efficiency in China, (3) suggest that after property rights reform, institutional rules for the water price, irrigation, and management systems need to be improved. The following institutional paths need to be focused on: the promotion of water-saving techniques by farmers; the maintenance of facilities; the water supply capacity of facilities; and farmers' water scarcity awareness, which leads to the improvement of irrigation efficiency.

*5.2. Discussion of the Differences in the Effects of Different Property Rights Models on the Irrigation Efficiency of Farm Households*

This study demonstrates that the "MCG-Model" has higher irrigation efficiency than the "private contract governance model". This is in contrast to the study by Chang (2022) [12], who concluded that irrigation efficiency is higher in the private management model than in the group management model. However, Chang's study was not conducted under the premise of clear property rights of farmland water facilities, and he defined the model in terms of who is the main body of irrigation facilities management and maintenance. However, the differences between the "MCG-Model" and the "PCG-Model" in the present study are not only the difference in the main body of management but also the differences in the water tariff system, irrigation system, and management system based on the different ownership of irrigation facilities. Therefore, this study is an advancement compared to the existing studies and explores the impact of the farmland water management model on the irrigation efficiency of farmers from a broader and deeper perspective. This study points out that different property rights models can lead to differences in the institutional pathways of farmers' adoption of water-saving technologies, facilities' water supply capacity, and ultimately to differences in irrigation efficiency.

*5.3. Further Discussion of the Advantages and Disadvantages of Two Models of Property Rights Governance in China's Yunnan Reform Pilot*

1.    Multiple Cooperative Governance Model

Advantages: Compared with the results of existing studies on the organizational form of farmland water governance [53,54], the advantage of this model is that it breaks through the institutional dilemma of farmland water facility management since the abolition of the "agricultural fee tax" and "two-worker system". That is, after the clarification of property rights, village collective organizations collect water charges to provide a source of funding for the maintenance of water facilities so that these facilities can continue to operate and be maintained. The emergence of the "village elite" as a water manager broke the previous dilemma of the lack of collective action leaving farmland water facilities unmanaged and unmaintained. Therefore, the operation of farmland water facilities and the provision of irrigation services by water stewards have re-established the relationship between grassroots organizations and farmers in terms of water use benefits, with grassroots organizations gaining collective economic income from water charges collection and farmers gaining the benefits of convenient water use, and the irrigation efficiency was also improved. At the same time, the MCG-Model has a wide range of implications as it contributes to the optimization and extension of the current rural grassroots self-governance system in China to a certain extent.

Disadvantages: The disadvantage of this model is that some farmers' interests are lost. Farmers are not able to reflect the role of rational economic man in irrigation water use. Regardless of whether irrigation water or water-saving technology is adopted, farmers passively accept village collective unified arrangements. According to the research and interviews, the yields and incomes of farmers, in general, were relatively lower under the "MCG-Model" than under the "PCG-Model".

2. Private contracting governance model

Advantages: On the one hand, the professionalization of irrigation services [19]. With the introduction of social capital to participate in the operation and maintenance of farmland water facilities, contractors have become professional irrigators, greatly enhancing the convenience of irrigation for farmers and breaking the dilemma of the difficult governance of farmland water facilities after the decline in the capacity of collective action in the countryside. On the other hand, farmers receive more benefits [55]. In this model, farmers take the initiative in irrigation water use; farmers can play the role of rational economic persons in production and management and can irrigate at any time according to their own plot and crop water needs, greatly enhancing the regularity of production water use. Finally, the increase in crop yield and farmer income is higher than that in the "MCG-Model". These results, which are consistent with the findings of existing studies [15,16], provide further evidence that privatizing the operation of water facilities can be an effective solution to rural water management problems.

Disadvantages: On the one hand, it uses relatively larger amounts of water because demand-based irrigation, metered water pricing systems, and sprinkler irrigation technology can drive farmers to use more water; these are consistent with existing studies [30,31], resulting in relatively inefficient irrigation for farmers. On the other hand, the emergence of contractors and the ability of contractors are uncertain. First, there is no guarantee that every village will have a contractor for these water facilities, and second, there is no guarantee that all contractors will have a responsible attitude to serve farmers. This instability may affect the interests of farmers and national food security, so it has yet to have a universal applicability in the short term.

*5.4. Discussion of the Degree of Generalizability of the Research Results*

First, Privatizing the property rights of public facilities can improve their management and operation [14,15,19], which is consistent with the findings of this study. For the current situation in China, clear property rights of agricultural water facilities should be universally implemented in the country. In addition, other countries or regions where the governance of agricultural water facilities is chaotic can also try to implement the reform of property rights of these facilities.

Second, considering the current problems of the rural grassroots organization system in China, it is feasible to implement the "MCG-Model" in China in general.

Finally, in China and other countries and regions with more developed rural economies and conditions for the market-based operation of farmland water facilities, the "PCG-Model" can be implemented according to local conditions.

*5.5. Shortcomings of the Study*

Due to the limitation of data volume, not all reform pilots are empirically analyzed except for the typical case pilots in this paper, which sheds light on the direction of further efforts of this group in future research on the property rights of Chinese farmland water facilities.

**6. Conclusions**

The findings of this study are as follows: Farmland water property rights reform in Yunnan, China, has significantly improved irrigation efficiency for farmers, and the "multiple cooperative governance model" has better irrigation efficiency than the "private contract governance model". The reform of property rights of farmland water facilities indirectly influenced the improvement of irrigation efficiency of farmers by promoting the adoption of water conservation techniques and the quality of facility maintenance; however, farmers' awareness of agricultural water scarcity decreased after the reform, leading to a decrease in farmers' irrigation efficiency. There are differences in the intrinsic governance logic between the MCG-Model and PCG-Model, which ultimately lead to the

former outperforming the latter in terms of irrigation efficiency. The findings of this paper have the following policy implications:

1.  China should continue to promote reform of property rights of farmland water facilities to effectively solve the existing problems of confusing management services and management of water facilities by "liberalizing ownership rights, activating operation rights, strengthening supervision rights, and clarifying revenue rights", at the same time to develop and improve the management systems, thus to promote the effective operation and maintenance of farmland water facilities.
2.  Based on the improvement of the system, we should promote the adoption of water-saving technologies by farmers, improve the quality of operation and maintenance of facilities, enhance the water supply capacity of facilities, and guide farmers to enhance their awareness of water conservation to promote water conservation.
3.  The "MCG-Model" should be implemented to solve the problem of confusion in the grassroots governance of farmland water facilities.
4.  The "PCG-Model" can be implemented in some areas that are suitable for the market operation of farmland water facilities in accordance with local conditions.

**Author Contributions:** Y.F.; writing the full text and analysis; M.C. and E.L.; methodology and model construction and data processing, J.L.; theoretical framework construction and research hypothesis. All authors have read and agreed to the published version of the manuscript.

**Funding:** This study was supported by the following grants, (1) National Natural Science Foundation of China (NSFC) Youth Project "Study on the influence mechanism and optimization strategy of farmland water management model on irrigation water efficiency of farmers": 7220030895; (2) National Key Research and Development Program, Intergovernmental Cooperation in International Science and Technology Innovation/Key Project of Hong Kong, Macao and Taiwan Science and Technology Innovation Cooperation, "Sino-Thai Cooperation on Community Water Management for Climate Change Adaptation": 2017YFE0133000; (3) Chinese Academy of Agricultural Sciences Science and Technology Innovation Project: 10-IAED-06-2023.

**Institutional Review Board Statement:** Not applicable for studies not involving humans or animals.

**Data Availability Statement:** No new data were created or analyzed in this study. Data sharing is not applicable to this article.

**Conflicts of Interest:** The authors declare no conflict of interest.

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
