# Peer review of "Has Property Rights Reform of China’s Farmland Water Facilities Improved Farmers’ Irrigation Efficiency?—Evidence from a Typical Reform Pilot in China’s Yunnan Province"

_agriculture, doi:10.3390/agriculture13020275_

Round 1

Reviewer 1 Report

- Keywords should be modified and introduced country (or region) (i.e:  Property rights reform of farmland water conservancy facilities is too long). 

-Introduction is lengthy. Needs to be shortened

- More details are needed about the survey conducted and the sample used (distribution per size, etc.......) . The authors also should indicate what extent can the results be generalized?

- The authors should explain the reasons of the methods used in the empirical analysis .  For example Tobit model is used not for the fraction of observations that takes the value of zero as indicated in line 392 but for the value of one. In fact, max of WUE (water efficieny) is one and not value zero.  Why PSM method is used instead of difference in difference method for example?.

- The paper requires some language revision and better editng.

- Please use some abbreviations in the text. For example,  WUAs for water users associations, WST for water-saving technology, etc...

Author Response

Response to Reviewer 1 Comments

First of all, on behalf of all authors, I would like to thank the reviewer for your hard work and we have made the following changes based on the reviewer's comments.

  1. we have shortened the length of the keywords and added country and regional keywords.
  2. we have streamlined and edited the introduction to shorten its length.
  3. In section 3.2, we added a detailed description of the survey and use of the sample, as well as a description of the statistical distribution.
  4. And added section 5.4, which provides a discussion of the extent to which the results of this study can be generalized.
  5. we have provided a description of why the Tobit model and the PSM method were used and the reasons why.
  6. We trimmed the language of the article, modified and optimized the language details.
  7. we have abbreviated the article based on reviewers' comments, e.g., abbreviating Water Conservation Technology Adoption (WATS), Water Users' Associations (WUAs), Multi-Cooperative Governance Model (MCG-Model), and Private Contracting Governance Model (PCG-Model).

Reviewer 2 Report

This is a very interesting study with important implications regarding the difficult subject of agricultural water resource management in a large, dynamic country. The interpretation of socio-economic findings is complicated, and different people could reasonably interpret the same results in slightly different ways. However, I believe that authors have done a great job in presenting complex data and came to reasonable conclusions supported by these data.

Please find my suggested edits in the attached file. Some of my comments may not be appropriate as I may have misunderstood what was intended or system being describe. If so, please disregard. 

Author Response

Response to Reviewer 2 Comments

      First of all, on behalf of all authors, I would like to thank the reviewer for your hard work and we have made the following changes based on the reviewer's comments.

  1. we optimized and revised the text, punctuation, capitalization, paragraphs, and other details in the full manuscript.
  2. we adjusted the content inside Figure 1 based on the reviewers' comments.
  3. we replaced Figures 2 and 3 and added descriptions of the graphic frames to better convey the meaning of the figure legends.
  4. in section 3.1, references were added to the data processing section
  5. in section 3.2.1, we limited the irrigation efficiency to nominal irrigation efficiency
  6. we corrected the confusion of two variable names with two rows of data that appeared in the data of descriptive statistics of variables in Table 3.
  7. We modified the second part of Section 5.3 by changing the sentence expression to be more closely related to the study results.
  8. We have further optimized the discussion section by adding section 5.4, which discusses the scope of application of the study results.
  9. Based on the reviewers' suggestions, we deleted the second clause of the policy revelation in the conclusion section and optimized the study conclusion.
